# Japanese Honeybees (*Apis cerana japonica* Radoszkowski, 1877) May Be Resilient to Land Use Change

**DOI:** 10.3390/insects12080685

**Published:** 2021-07-30

**Authors:** Philip Donkersley, Lucy Covell, Takahiro Ota

**Affiliations:** 1Lancaster Environment Centre, Lancaster University, Lancaster LA1 4YQ, UK; lucy.covell@gmail.com; 2Graduate School of Fisheries and Environmental Science, Nagasaki University, Nagasaki 852-8521, Japan; takahiro@nagasaki-u.ac.jp

**Keywords:** pollinator, landscape, land use, urban rural gradient, Japanese honeybee, honey, pollen, nutrition

## Abstract

**Simple Summary:**

Pollinators are threatened globally by growing urban sprawl and agriculture. The Western Honeybee (*Apis mellifera*) readily adapts to whatever food is available, so people have made it the most widely distributed pollinator across the world. Previous research has suggested that the Western Honeybee may be less resilient to land use change outside of its natural range. This study examines a different honeybee species—the Japanese Honeybee (*Apis cerana japonica*). Unlike the Western Honeybee, this species is found almost exclusively in its natural range in Japan. Consequently, it may be better adapted to its local food sources and therefore more resilient. Working in southern Japan, in the Nagasaki and Saga prefectures, we looked at the nectar and pollen that the Japanese Honeybee feeds on. Their food intake was then examined in relation to local land use composition. We found minimal impact of increasing urban sprawl on the forage of the Japanese Honeybee. This goes against previous studies on the Western Honeybee elsewhere in the world. Though in need of a direct comparison with Western Honeybee, these preliminary results could be due to differences in urban green infrastructure in Japan, or due to an adaptation by the Japanese honeybee to its surroundings.

**Abstract:**

Pollinators are being threatened globally by urbanisation and agricultural intensification, driven by a growing human population. Understanding these impacts on landscapes and pollinators is critical to ensuring a robust pollination system. Remote sensing data on land use attributes have previously linked honeybee nutrition to land use in the Western Honeybee (*Apis mellifera* L.). Here, we instead focus on the less commonly studied *Apis cerana japonica*—the Japanese Honeybee. Our study presents preliminary data comparing forage (honey and pollen) with land use across a rural-urban gradient from 22 sites in Kyushu, southern Japan. Honey samples were collected from hives between June 2018 and August 2019. Pollen were collected and biotyped from hives in urban and rural locations (n = 4). Previous studies of honey show substantial variation in monosaccharide content. Our analysis of *A. cerana japonica* honey found very little variation in glucose and fructose (which accounted for 97% of monosaccharides), despite substantial differences in surrounding forage composition. As expected, we observed temporal variation in pollen foraged by *A. cerana japonica*, likely dependent on flowering phenology. These preliminary results suggest that the forage and nutrition of *A. cerana japonica* may not be negatively affected by urban land use. This highlights the need for further comparative studies between *A. cerana japonica* and *A. mellifera* as it could suggest a resilience in pollinators foraging in their native range.

## 1. Introduction

Globally, land use change is driving biodiversity loss. Primarily driven by anthropogenic uses, such as agricultural intensification and urban development, these losses threaten ecosystem services, and impact the growing human population [1,2]. One such ecosystem service is provided by insect pollinators. Pollinator declines are linked to a reduction in both nesting sites (for wild bees) and available forage (for all pollinators) [3]. Land use change has damaged the availability and variety of pollen [4,5]. This is particularly evident in specialist pollinators, which can only utilise certain suitable flora [6,7]. Consequently, this is particularly important for specialist pollinator species that are under selection pressure to make a transition to newly available sources of food [5,8].

In a landscape context, pollinator activity shifts depending on the pollinator species’ forage preferences, and the nutritional value of the pollen [9,10,11,12,13]. Pollinator health and ecosystem service provision may effectively be determined by land use change, in combination with other significant factors [14,15,16]. These effects have only recently begun to be incorporated into the conservation literature, where papers highlight the importance of landscape heterogeneity, trees, matrix effects, habitat loss and fragmentation on pollinator diversity and success [17,18,19].

Amongst all animal pollinators, bees in particular have been affected on a global scale by land use change [19,20,21,22]. Western Honeybees (*Apis mellifera L.*) remain a model species of understanding pollinator health, nutrition, behaviour [23,24,25] and are key organisms in the development of mathematical algorithms and behavioural models for pollinators [26,27,28], and as such are still used an important model species for studying insect pollinators globally. Yet, this simplicity may obscure the importance of considering all pollinator species and the impacts of these factors on wild pollinators. The behavioural, ecological and evolutionary differences between *A. mellifera* and other genera of insect pollinators, for example: pseudo-social bees (e.g., *Bombus terrestris*) or solitary bees (e.g., *Osmia bicornis*) are vast. In this study, we examine substantial differences within the genus *Apis*, further highlighting the inadequacy of *A. mellifera* as a “catch-all” species for pollinator decline.

In Japan, both *A. mellifera* (the western honeybee) and *Apis cerana japonica* (the Japanese honeybee) are managed for their pollination, honey production and cultural/heritage values. Arguably, elsewhere in the world the Western honeybee has become one of several factors negatively impacting the health of native, wild pollinators. Tatsuno and Osawa [29] found that the native Japanese honeybee pollinate more native species and more species overall, making them potentially one of the more important pollinators in the country. Despite this, Western honeybee are dominant in the Japanese beekeeping industry [30], their widespread use, equivocal with “livestock” management, highlights the importance of examining how the Western honeybee may be impacting other native pollinators, like the Japanese honeybee.

### 1.1. Japanese Beekeeping

Western honeybees were originally introduced to Japan as they have a higher honey production and lower swarming rate than Japanese honeybees. There was a significant decline in the prevalence of native beekeeping in Japan until 2005, determined by an increasing reliance on imported honey (more than ten times greater than domestic supplies) [30]. This was not only due to a decline in beekeeping as a profession, but also a decline in nectar sources, especially orange trees [30]. This reduction in pollinators has reduced the pollination services provided to cultivated crops, thus negatively affecting yield and quality of produce [30].

This decline has now stabilised, due to increases in urban and small-scale beekeeping [31]. The Japanese Government amended the Apiculture Promotion Act in 2012, which required hobbyist beekeepers to report their number of hives. The most recent statistics available estimates 10,000 active beekeepers in Japan, growing from approximately 2000 when the Apiculture Promotion Act was introduced in 1995 [32]. For context, the National Bee Unit in England estimates the number of beekeepers at 44,000, a number that has remained relatively stable over the past 5 years [33].

Local expert knowledge from beekeepers is key to understanding the behaviour and adaptations of their bees. Beekeepers in the Nagasaki prefecture have said that the Western honeybee has adapted to forage more on “mass-flowering plants” than the Japanese honeybee. Beekeepers and academics have observed that the Japanese honeybee instead forages on more diverse flower sources, collecting nectar and pollen from anything available, rather than through the “flower constancy” behaviour observed in Western honeybee [34,35]. The beekeepers believe that “if nectar is very scarce (like in urban area), *cerana* can more successfully find small patches on which to survive (maybe in a small garden)” [36]. Observations by this group comparing Western honeybee and Japanese honeybee when kept in the same apiary have led the beekeepers to believe that “*mellifera* is not good at collecting nectar in summer compared with *cerana*”, especially in urban environments with less spatially extensive flower patches [36]. It has recently been identified that Western honeybee are more susceptible than Japanese honeybee to predation from various native hornets in Japan [29,30].

Beekeeping is growing in Japan, as more people take it up as a hobby or a business interest. Investigating the threats to pollinators and their ecosystem services are equally increasing in importance. Identifying land uses and plant species that support *A. cerana japonica* may be key in maintaining the success of both urban and rural commercial and hobbyist beekeepers, as their numbers continue to grow.

### 1.2. Aims and Scope

The sugar composition of honey is directly linked to the flowers that have been foraged by bees to make it. Phytochemicals derived from the flowers convey unique biochemical properties to honey, which has led to marketization of particular honeys for their medicinal properties (such as “Manuka”) [37,38,39]. The glucose/fructose ratio of honey can impact its palatability for humans [40,41], and higher glucose levels contribute to the production of hydrogen peroxide, an important antimicrobial compound found in honey [42,43].

This study aims to investigate pollen and nectar foraging by *A. cerana japonica*, studying urban and rural populations maintained by hobbyists in Nagasaki and Saga prefectures, on Japan’s southern island of Kyushu. Here, beekeeping is practiced by a small beekeeping community, largely producing honey part-time for personal use and sale, with a few full-time commercial producers. *A. cerana japonica* are almost exclusively chosen by hobbyist beekeepers in Nagasaki (though *A. mellifera* continue to be favoured by commercial beekeepers). The factors affecting pollinator health and honey production in Japan are not well studied, so this study focuses on answering the two following key questions:How does land use in Japan affect the honey produced by *A. cerana japonica*?How does the time of year and location affect the pollen collected and honey produced by *A. cerana japonica*?

## 2. Materials and Methods

### 2.1. Honey Sampling

Honey was collected from 22 hives across Nagasaki and Saga (Figure 1) between June 2018 and August 2019. A questionnaire was given to the owner of each hive to determine information such as the species of bee, the location of their hive and the environment surrounding the hive (Appendix A).

Honey samples were analysed using high-performance liquid chromatography (HPLC) to determine sugar composition (fructose, glucose and maltose), following methods for honey analysis used in Ouchemoukh et al. [44]. These sugars were selected due to previously observed geographic variation in composition [45].

Crystallised honey samples were heated in a 90 °C water bath until clear and left to cool to room temperature. All samples were then diluted using High Performance Liquid Chromatography (HPLC)-grade water to a ratio of 1 µL honey per 100 mL water. Two repeats of 10 µL solution were run through the HPLC, testing for fructose, glucose and maltose levels. Three repeats per sample were conducted and the means taken.

Aliquots were analysed on an Agilent Analytical 1200LC HPLC machine (Agilent Systems, UK) using a Thermo Dionex CarboPac PA20 Analytical column, 3 × 150 mm (Thermofisher, UK). A Pulsed Amperometric Detection (PAD) detector was used, and samples were transported in HPLC grade water/200 nM NaOH. Detection peaks were quantified against a dilution series of standards for fructose, glucose and maltose.

### 2.2. Pollen Sampling

Pollen samples were collected at least once every three weeks for nine weeks from 13.6.2019 to 8.8.2019 at two urban (Hives 15, 16) and two rural hives (Hives 11, 19) in Nagasaki-ken. Pollen was sampled via pollen trapping, direct pollen basket collection and brood chamber sampling, as these methods have previously been used to sample pollen successfully [46,47]. Samples at all hives were collected from the brood chamber, as this was found to be the quickest and most effective method, as well as arguably the least disruptive. Typhoons during weeks four and seven limited sampling during these periods; one hive was abandoned by the colony in week eight (detailed information available in Appendix A).

Biotyping of pollen grains was used to provide a descriptive measure of the diversity of foraged pollen in samples [48]. Pollen samples were purified via acetolysis using methods based on Jones [49]; each sample was imaged five times with a digital microscope. Pollen grains were then counted and identified into biotypes based on physical properties (Appendix A), due to low sample site replication, no further analysis was performed on the pollen data.

### 2.3. Land Use Composition

To analyse the correlation between land cover and Japanese honeybee nutrition, data were sourced from the Japanese Ministry of the Environment’s Biodiversity Centre (Appendix A). The composition and configuration of different land uses in the surrounding 1, 3 and 5 km radii of each hive were measured using the *radius* tool in ArcGIS 10.8.1 (ESRI, US).

Radii of 1, 3 and 5 km were chosen as these cover the range of foraging distances travelled by bees from their hive [47]. Land cover classes that accounted for <0.5% of total cover within a buffer zone were excluded from analysis. The dominant land use (the land use contributing the greatest percentage of land cover) and the ratio of urban-to-rural land uses were then calculated using these data. The distance from each hive to the nearest urban area was also calculated based on methods established in Clermont et al. [50].

### 2.4. Statistical Analysis

Honey composition data were analysed to determine inter-hive variance in sugar content. Monosaccharide composition of the honey samples was analysed by Pearson’s correlation with three land use factors: distance to urban areas, dominant surrounding land use and ratio of rural-to-urban land use. Critical P-scaling was performed to control false discovery rate (FDR) on these multiple land use analyses [51], for these analyses critical P was set at α = 0.0115.

Pollen count data were not analysed due to low hive level replication. The number of observed biotypes (as a rough approximation of pollen species richness), and sample date and hive are presented in Appendix A. All other analyses were all carried out in *R* statistical software version 4.0.5 [52].

## 3. Results

### 3.1. Honey Sugar Composition

Honey samples from 22 different hives across Japan, were analysed for fructose, glucose and maltose content. Glucose accounted for the majority of mono-saccharides within the honey samples, accounting for 59.3 ± 0.4% (mean ± S.E.); followed by fructose at 38.1 ± 0.5%, and maltose sugars being present in trace amounts at 2.5 ± 0.2%. The sugar ratios of honey samples did not change significantly between hives and apiary location (H = 21, df = 21, *p* = 0.459; Figure 2).

Although fructose and glucose content remained consistent between hives, significant inter-hive variance was observed in maltose content (F = 0.465, df = 21, *p* = 0.029). The distance to the nearest urban area had no effect on the proportions of sugars in honey samples (*fructose*: r_s_ = −0.099, *n* = 22, *p* = 0.660; *glucose*: r_s_ = −0.284, *n* = 22, *p* = 0.200; *maltose*: r_s_ = −0.055, *n* = 22 *p* = 0.807).

The proportions of sugars found in the honey samples did not change depending on the date the sample was collected (*fructose*: H = 19.791, *n* = 22, *p* = 0.285; *glucose*: H = 20.146, *n* = 22, *p* = 0.267; *maltose*: H = 19.708, *n* = 22, *p* = 0.289).

### 3.2. Pollen Biotype Composition

In the process of biotyping, 262 microscope images were counted across 131 samples from four *A. cerana japonica* hives collected between June and September 2019. Across these samples, 50 biotypes were identified (Appendix A). Biotypes 1, 3, 10, 14, 17 and 20 had a total abundance greater than 1000 grains across all sample images, and thus were deemed ‘dominant’ biotypes. A description of these biotypes can be found in Table 1. Statistical analyses of pollen biotypes use these groups throughout, as well as ‘others’ (combined totals from biotypes 2, 4–9, 11–13, 15, 16, 18, 19 and 20–52).

Based on acetolysis, the most common species of pollen found within the study area were provisionally identified as Japanese angelica tree (*Aralia elata*), wutong (*Firmiana simplex*), Japanese Oak (*Lithocarpus edulis*), East Asian mallotus (*Mallotus japonicus*), kumquat (*Citrus/Fortunella crassifolia*), Tree Ivy (*Dendropanax trifidus*) and Japanese spindle (*Euonymus japonicus*).

### 3.3. Land Use Composition

Within a 1 km radius of the 22 hives, eight land types were dominant (Appendix A). Within a 3 km radius, this was six land types, and in a 5 km radius, this was five land types. On average, within a 1 km radius around each hive, rural land use was dominant, covering an average of 80.0 ± 6.7% of all land. This land cover varied between hives (Figure 3 and Appendix A), but this variation was not reflected in fructose or glucose content of honey. The maltose content of honey increased as rural land cover within 1 km radius of the hive increased, but not significantly under P-scaling to control FDR (Table 2).

## 4. Discussion

In this study, fructose and glucose sugars made up the majority (an average of 97.5 ± 0.9%) of the three monosaccharide sugars analysed from the honey of the Japanese honeybee. Our study was based on sugar analysis from sites across south Japan (*n* = 22), as well as pollen collections from a smaller number of sites (*n* = 4), sampled repeatedly (*n* = 13 per hive). Our results show minimal variance in the sugar composition across a wide geographical area, and a broad diversity of pollen types associated with these bees.

Despite the small replication size of pollen trapping, the sampling regime we attempted was able to show temporal variation in pollen forage composition. Flowering date is a significant factor impacting pollinator foraging [9,54,55], and our study was no exception. As a result, it is important to focus on ensuring a consistent provision of pollen sources throughout the year, by maintaining heterogeneity of flowering times within vegetation cover [12,48].

### 4.1. The Importance of Honey Sugar Composition

The biochemical properties (i.e., antibacterial activity) of honey varies according to its sources, as do its physical and chemical properties [38,39,56]. These properties are as important to humans for their medical properties as they are to the honeybees for their potential to act as a “bee pharmacy” [57,58]. Variation in these properties has been suggested to be part of a suite of factors negatively impacting honeybee health globally [16,59].

Consequently, flower resources (availability and botanical origin) in the urban environment and subsequent variance in honey sugar composition have the potential to substantially impact bee health. Previous research in the Western honeybee from western urban-rural gradient landscapes has shown that increasing urban land use has a negative impact on the nutrition of these eusocial bees [9,10,11,12,13]. Yet, in our study measures of land use composition, for example rural-to-urban ratio or distance to urban landscapes, did not significantly impact sugar composition. Maltose levels appeared to vary substantially more between hives. Maltose is often added to man-made products, as it is not commonly naturally occurring [60]. However, studies have shown that the sugar composition of honey changes over time, due to various chemical processes enabled by the heat of hives [40,61]. This can lead to the production of maltose from glucose and fructose, meaning that the maltose may have formed during storage [61].

No link between nutrition of the Japanese honeybee and the composition of their environment suggests that Japanese honeybees in Japan may not be perturbed by the negative effects of urbanisation observed in other studies of honeybees. The question remains, as to why: is it land use being less detrimental, or are Japanese honeybee being more resilient to land use change?

### 4.2. Could Urban Land Use Be Less Detrimental?

As observed by beekeepers involved in our study, when comparing Western honeybee with Japanese honeybee, the latter may prefer to forage on a wider array of available forage [36]. The low variance in monosaccharide composition in our study of Japanese honeybee, compared with the significant inter-hive variance in sugar composition from studies of Western honeybee [45], suggests Japanese honeybee may be better at regulating the balance of sugars in their nectar forage intake.

One could suggest this is due to differences in the landscape composition impacting the bees. Within the context of our study (Nagasaki and Saga), the rural-urban matrix of landscapes was significantly variable between sites (Appendix A). Our findings may show that Japanese honeybee could be successfully exploiting the diverse nectar sources available to them in these environments. This may suggest that Japanese urban landscapes are not as detrimental to Japanese honeybee as western urban environments are to Western honeybee, as found in existing studies that have explored the impact of urban environments on pollen forage and pollinator health [50,62,63]. Few studies have explicitly studied the link between landscape “quality” and nectar foraging as has been attempted in our study. Homogenous landscapes (i.e., agricultural monocultures) have previously been shown to negatively impact the nectar foraging of Western honeybees [64]. The effect of urban environments on nectar foraging remains largely unexplored for Western honeybees however. Furthermore, there is a growing literature on urban green infrastructure that suggests cultural and demographic factors influence the distribution and composition of urban green spaces [65,66].

Elsewhere in the world, where urban land uses dominate, the managed floral composition and overall lack of availability of nectar has been shown to have detrimental effects on honeybee nutrition [50,63]. Previous studies directly comparing the effects of land use composition on honeybee nutrition found significant effects on pollinator foraging and nutrition [9,67,68]. The lack of impact from landscape composition on nutrition of the Japanese honeybee here suggests urban flora may not be perturbing foraging by Japanese honeybee. Cultural, historic and heritage factors that impact the composition of natural elements in urban settings may explain these differences. If urban green spaces are providing a stable and sufficient nectar source for Japanese honeybee [65,69], prefectures like Nagasaki and Saga may become important case studies in urban landscapes that support local bee populations.

The social construct of what constitutes “urban” in Japan is observably distinct to other sites in which pollinator ecology has been researched, in terms of extent and age of urban green spaces, the social expectation for preserving and cultivating these spaces, as well as social engagement with natural history. [65,70,71,72]. In previous studies, urban environments are often dominated by invasive or alien species, supplanting native flora [73,74]. Cultural factors leading to an emphasis on native flora [65,71], as well as strict plant quarantine and biosecurity measures [75], may have contributed to urban environments in Japan being more “native” than western urbanised landscapes. Emergent analyses of urban green infrastructure suggest remote sensing data may bear out this comparison [65,66,76], but a direct comparative analysis with sufficiently controlled parameters is currently lacking in the literature.

### 4.3. Could Japanese Honeybees Be More Resilient?

The other driver of our observed minimal disturbance of Japanese honeybee within urban environments and agricultural rural environments is that Japanese honeybee itself may be better adapted to these environments. Due to the lack of significant correlation with land use composition on honeybee nutrition, we suggest that in Nagasaki and Saga, Japanese honeybee may in fact be “coping” with the presence and expansion of urbanisation. Previous studies by Garbuzov et al. [69] and Lowenstein et al. [8] suggest that pollinators can adapt successfully to urban living. However, notably, due to a lack of direct comparison with Western honeybee or experiments relocating Japanese honeybee to a non-localised environment, determining this link remains speculative.

Some studies suggested that honeybees may prefer key plant species because of their ability to perceive higher nutritional value in these plants [77,78]. We suggest that it is possible the Japanese honeybee may have co-evolved with their native flora to have these advantageous preferences. Whereas the Western honeybee, being both a global generalist and alien species to this location, is not connected with the local flora via eco-evolutionary interactions and therefore is lacking this foraging and nutritional advantage.

When comparing the biology of Western honeybee with Japanese honeybee, studies of the “waggle dance” properties suggest that the flight capacity of Western honeybee may be up to two times larger than Japanese honeybee [35,79]. The suggestion, that different species of honeybee might possess distinct ‘dialects’ of the waggle dance, remains controversial. Direct comparisons of waggle dance properties between these two species suggest that their “dialects” (i.e., the forage distance-to-dance duration) scale differently [29,80]. Foraging over larger distances is typically advantageous to generalist foragers like the Western honeybee, though it suggests that Japanese honeybee may be better adapted to smaller areas with forage sources that are more difficult to locate [80].

Denser, more homogeneous and/or larger urban areas may have a significant effect on Japanese honeybee [54]. The area studied in this paper did not have many high population density (Nagasaki: 335 people/km^2^, Saga: 342 people/km^2^) or large urban areas, so this may be a relevant factor to incorporate in future studies. However, based on the results of this study alone, we found that the proximity of hives to urban areas is not an issue for regulation of honey composition, which means that the Japanese honeybee may be resilient in increasingly urbanised areas. This contradicts research on Western honeybee, which has highlighted the threats of urban and agricultural expansion. Here, we may be observing a difference between Western honeybee and Japanese honeybee, a resilience to land use variance in Japanese honeybee not found in other honeybee species [12,63]. One key factor when considering local environmental impacts on pollinators is activity outside of their native range. Whereas Western honeybee has a cosmopolitan distribution, well outside of its native range [81], Japanese honeybee is almost exclusively present within its native range in Japan. Any apparent resilience of this latter species may be linked with this, highlighting it as an important factor in future studies of pollinator resilience relative to its range/distribution.

## 5. Conclusions

Our study found that the sugar composition of honeys collected by Japanese honeybee does not vary significantly between hives across a wide geographical area. Furthermore, no significant correlation was found with land use composition surrounding each hive. Our understanding of the importance of sugar composition for hive health, the impact of botanical origins on sugar composition, and local knowledge on the foraging behaviour of Japanese honeybee leads us to conclude that this particular bee species may be sufficiently able to adapt to local land use conditions. By this, we mean the Japanese honeybee honey forage is not negatively affected by land use, as has been seen in studies of the Western honeybee [64].

Comparing Japanese honeybee to Western honeybee would potentially identify factors allowing the Japanese honeybee to successfully exploit both urban and rural environments in Japan, when the Western honeybee tends to be less successful across the planet. It would also be beneficial to identify local plant species abundances relative to the pollen spectrum foraged by the Japanese honeybee, to maximise benefits of any vegetation conservation efforts.

To discover more of the effects of various external factors on honey composition, more complex compositional analyses, including other sugars such as melezitose and sucrose, as well as other components such as pollen grains could be relevant [44,82,83]. In summary, our study has provided a first-look at the nutrition of *A. cerana japonica* in its native habitat, which suggests these bees in particular may be resilient to the effects of land use change that have negatively affected other honeybee species elsewhere in the world.

## Figures and Tables

**Figure 1 insects-12-00685-f001:**
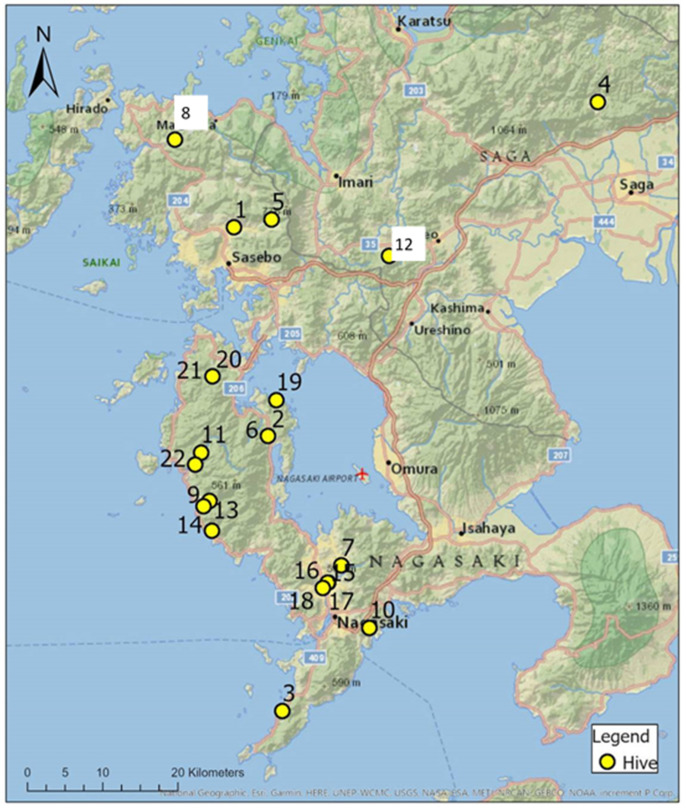
Locations of 22 hives from which honey samples were collected, clustered hive locations (Hives 7, 15–18) are also available in Appendix A.

**Figure 2 insects-12-00685-f002:**
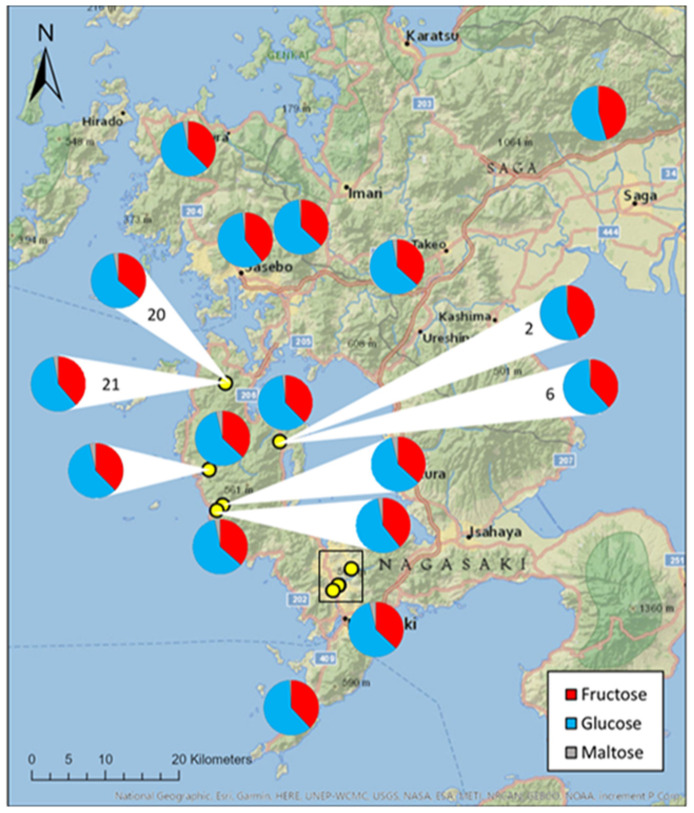
Proportion of fructose, maltose and glucose in honey samples collected from 22 *Apis* species hives across Kyushu, Japan. Yellow dots represent hives where pie chart could not be placed directly in the correct location. Where multiple hives had the same location, pie charts are labelled with hive number. Sugar proportions were calculated using HPLC. Clustered hives located in Nagasaki, within the black rectangle, can be found in Appendix A.

**Figure 3 insects-12-00685-f003:**
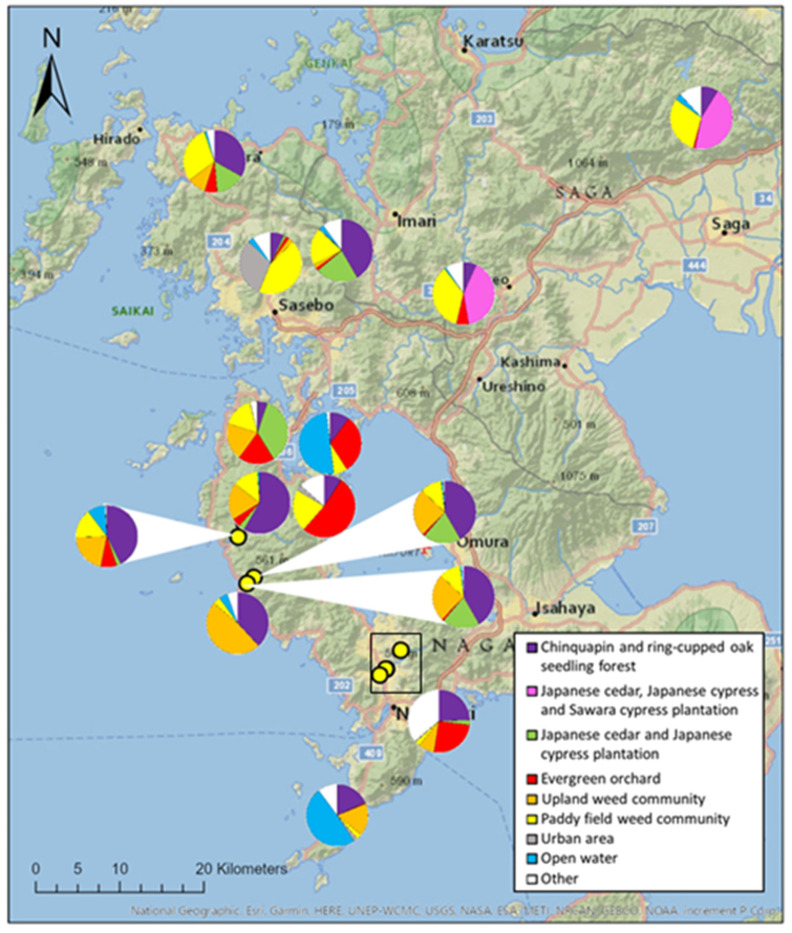
Proportion of land uses in a 1 km radius surrounding 17 hives across Kyushu, Japan. Land uses which were dominant for at least one hive are shown, along with all other land types, grouped into ‘other’. Yellow dots represent hives where pie chart could not be placed directly in the correct location. Hives 2 and 6 as well as 20 and 21 were located in the same apiaries, so are shown using one pie chart per location. Clustered hives in Nagasaki, within the black rectangle, can be found in Appendix A. Land use was calculated using vegetation maps freely available from the Japanese Ministry of the Environment’s Biodiversity Centre.

**Table 1 insects-12-00685-t001:** Dominant biotype classifications and descriptions used to identify pollen grains found in samples collected from five *Apis cerana japonica* hives in Japan, between June and September 2019. Potential family and species names are also given where possible, based on pollen samples collected directly from flowering plants and identified using a palynomorph guide of Japanese flora by Shimakura [53]. Reference images are approximately relatively sized, as indicated by scale bar (50µm).

Biotype	Possible Species and Family Name (s)	Description	Reference Image
1	*Aralia elata* (Araliaceae)	Small, circular/semi-circular	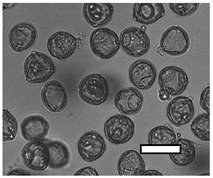
3	*Firmiana simplex* (Malvaceae)	Medium, dark, 3-way symmetry, rounded	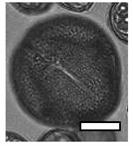
10		Very small, light, circular/semi-circular.	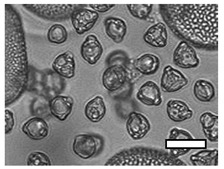
14		Medium, oblong, slightly pointed at ends, arcing lines through	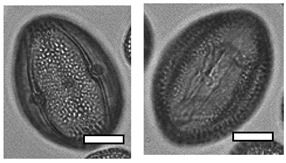
17	*Lithocarpus edulis* (Fagaceae)	Small, oval, lines arcing through.	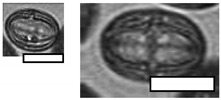
20	*Mallotus japonicus* (Euphorbiaceae) *Citrus*/*Fortunella crassifolia* (Rutaceae) *Dendropanax trifidus* (Arialaceae) *Euonymus japonicus* (Celastraceae)	Similar to 16, but more indents, elongated. From end on: small-medium, three rounded sides, triangle inside, with points between indents.	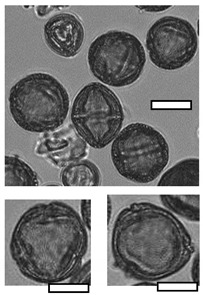

**Table 2 insects-12-00685-t002:** Statistical analysis of the effect of rural-to-urban ratio of land use on composition of sugars found in honey collected by 22 colonies of Japanese honeybee in various locations within Kyushu, Japan, between June 2018 and August 2019. Statistical values for rural-to-urban ratios within radii of 1, 3 and 5 km were calculated. N for all statistics was 22. All maltose values were calculated using Pearson’s product-moment correlation (r) and all fructose and glucose values were calculated using Spearman’s rank correlation (r_s_).

Sugar	1 km		3 km		5 km	
	r_s/_r Value	*p*-Value	r_s/_r Value	*p*-Value	r_s/_r Value	*p*-Value
Fructose	-0.230	0.303	0.079	0.726	0.174	0.439
Glucose	−0.095	0.675	−0.051	0.822	−0.297	0.179
Maltose	0.465	0.029	0.189	0.399	0.043	0.848

## Data Availability

All data generated or analysed during this study are included in this published article and its Appendix A.

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
