# Peer review of "Japanese Honeybees (Apis cerana japonica Radoszkowski, 1877) May Be Resilient to Land Use Change"

_insects, 2021, doi:10.3390/insects12080685_

Round 1

Reviewer 1 Report

The Authors have addressed adequately the comments of the first review. There are only some extra editing points, which might improve presentation and facilitate reading. Please see the attached file.

Author Response

The Authors have addressed adequately the comments of the first review. There are only some extra editing points, which might improve presentation and facilitate reading. Please see the attached file.

 *** We thank the reviewer for agreeing to review our manuscript a further time. We have agreed to each of the editing points the reviewer has recommended. For clarity, we have documented these changes below.

L254-259: We agree with the reviewer, this paragraph has now been moved to the introduction.

L264: Agreed, we have deleted this sentence.

L266: Agreed, we have edited this sentence in accordance with the reviewers recommendation.

L267-268: Agreed, we have deleted this sentence.

L270-271: Agreed, we have moved this sentence.

L272: Agreed, we have deleted this sentence.

L278-292: Agreed, we have edited and moved this sentence in accordance with the reviewers recommendation.

L293: Agreed and amended.

Reviewer 2 Report

Most of the issues raised by the reviewers were considered by the authors and this version of the manuscript is improved. However, further improvement is needed (details below), therefore I suggest major revision of the manuscript.

The authors insist that managed honey bees that are part of agricultural system may be studied within the context of pollinator decline that applies to wild pollinators only. I cannot recommend for publication manuscript equating the livestock (managed honey bees) with wild animals facing specific threats (pollinators facing pollinator decline). It should be clearly indicated by the authors that pollinator decline does not concern managed honey bees. The authors should also consider previous comments raised by the reviewers regarding this issue.

In lines 52-54 the authors refer to pollinator fitness mixing definition of solitary organism fitness with its specific version related to social organism. Please avoid such simplifications and please use evolutionary correct definition of fitness in the manuscript.

Please do not use abbreviations (WHB and JHB) in the manuscript. It makes the text hard to understand.

Assertion that western honey bee remain an important model species for studying insect pollinator dynamics globally (lines 58-59) is unfounded. Cited papers consider bees not all pollinators and even regarding bees such a strong statement cannot be provided.

I don’t understand the reason for providing the paragraph in lines 64-70. It is disconnected from the rest of the manuscript and impedes understanding of the text. I suggest removing it.  

Figure 1 – numbers 8, 12 and 15 are hard to read.

Assertion provided in lines 242-244 is risky and should not appear in the manuscript. What the authors call “high level of depth” in fact is high number of pseudoreplications. I suggest removing this kind of reasoning from the manuscript.

I don’t understand the point of the paragraph provided in lines 261-264. Why is this information provided and how it relates to the rest of the manuscript?

Discussion provided in lines 266-283 suggests that Japanese honey bees could simply prefer specific plant species when foraging, however this possibility is not discussed directly. I suggest addressing this possibility directly in the text. Some studies suggested that bees may prefer key plant species that are more nutritious than others. Is it possible that Japanese honey bees that co-evolved with the surrounding flora have such preferences, whereas western honey bee – alien species that is not connected with this flora via eco-evolutionary interactions – have not such preferences?

Please remove from the manuscript superfluous words. Why “immediately” is needed in line 290?

Please review studies cited in line 308. The sentence provided in lines 306-308 is not supported by studies 66-68 and studies 9 and 40 are self-citations.

Last sentence of the manuscript should provide take home message summing-up the current study without referring to any previous studies.

Author Response

Most of the issues raised by the reviewers were considered by the authors and this version of the manuscript is improved. However, further improvement is needed (details below), therefore I suggest major revision of the manuscript.

*** We courteously thank the reviewer for their comments on our manuscript. We have followed the reviewers comments and made the recommended improvements to the manuscript. Our responses are detailed below and marked in accordance with our previous response to reviewers comments (***).

The authors insist that managed honey bees that are part of agricultural system may be studied within the context of pollinator decline that applies to wild pollinators only. I cannot recommend for publication manuscript equating the livestock (managed honey bees) with wild animals facing specific threats (pollinators facing pollinator decline). It should be clearly indicated by the authors that pollinator decline does not concern managed honey bees. The authors should also consider previous comments raised by the reviewers regarding this issue.

*** We apologise to the reviewer that we were not clear enough in our phrasing of our revised manuscript. We have amended the manuscript substantially in this aspect, clearly highlighting that the western honeybee is not equal with wild pollinators in terms of the effects of global decline (indeed, arguably, they may be the cause of global declines of wild pollinators). We highlight that the results of studies on the impact of land use change on the western honeybee have previously demonstrated significant negative impacts on honeybee health.

In lines 52-54 the authors refer to pollinator fitness mixing definition of solitary organism fitness with its specific version related to social organism. Please avoid such simplifications and please use evolutionary correct definition of fitness in the manuscript.

*** Agreed, we believe we more accurately mean the “health” of this species and its capacity to maintain ecosystem services, not the reproductive capacity. We have amended as such throughout.

Please do not use abbreviations (WHB and JHB) in the manuscript. It makes the text hard to understand.

*** Agreed and amended throughout.

Assertion that western honey bee remain an important model species for studying insect pollinator dynamics globally (lines 58-59) is unfounded. Cited papers consider bees not all pollinators and even regarding bees such a strong statement cannot be provided.

*** We agree that this point requires a more subtle distinction, we maintain that the global effort to use Apis mellifera as a model species of understanding pollinator health, nutrition, behaviour and the development of mathematical algorithms and behavioural models based on A. mellifera emphasise its importance as a model species. And yet, the scientific consensus is that honeybees do not adequately represent all insect pollinators. We have amended the text to emphasise this point. It now reads: “Amongst all animal pollinators, bees in particular have been affected on a global scale by land use change [19–22]. Western Honeybees (Apis mellifera L.) remain a model species of understanding pollinator health, nutrition, behaviour [23–25] and are key or-ganisms in the development of mathematical algorithms and behavioural models for pol-linators [26–28], and as such are still used an important model species for studying insect pollinators globally. Yet, this simplicity may obscure the importance of considering all pollinator species and the impacts of these factors on wild pollinators. The behavioural, ecological and evolutionary differences between A. mellifera and other genera of insect pol-linators, for example: pseudo-social bees (e.g. Bombus terrestris) or solitary bees (e.g. Osmia bicornis) are vast. In this study, we examine substantial differences within the genus Apis, further highlighting the inadequacy of A. mellifera as a “catch-all” species for pollinator decline.”

I don’t understand the reason for providing the paragraph in lines 64-70. It is disconnected from the rest of the manuscript and impedes understanding of the text. I suggest removing it.  

*** Agreed we have shortened this paragraph to clarify its meaning. It now reads: “WHBs were originally introduced to Japan as they have a higher honey production and lower swarming rate than JHBs.”

Figure 1 – numbers 8, 12 and 15 are hard to read.

*** Agreed, we have now amended the figure to have white background on numbers 8 and 12. And the clustered hive numbers that are more clearly indicated in supplementary materials are included in the figure legend.

Assertion provided in lines 242-244 is risky and should not appear in the manuscript. What the authors call “high level of depth” in fact is high number of pseudoreplications. I suggest removing this kind of reasoning from the manuscript.

*** Agreed, though we have retained the account of our sampling depth in more neutral language as we feel it is an important caveat to be considered when interpreting the results of our study. It now reads: “Our study was based on sugar analysis from sites across south Japan (n = 22), as well as pollen collections from a smaller number of sites (n = 4), sampled repeatedly (n = 13 per hive)”.

I don’t understand the point of the paragraph provided in lines 261-264. Why is this information provided and how it relates to the rest of the manuscript?

*** Following this comment and edits recommended by other reviewers, this paragraph has now been amended to include an explanation as to why the biochemical properties of nectar are important considerations within previous models trying to understand honeybee health. It now reads: “The biochemical properties (i.e. antibacterial activity) of honey varies according to its sources, as do its physical and chemical properties [38,39,56]. These properties are as im-portant to humans for their medical properties as they are to the honeybees for their po-tential to act as a “bee pharmacy” [57,58]. Variation in these properties has been suggested to be part of a suite of factors negatively impacting honeybee health globally [16,59].”

Discussion provided in lines 266-283 suggests that Japanese honey bees could simply prefer specific plant species when foraging, however this possibility is not discussed directly. I suggest addressing this possibility directly in the text. Some studies suggested that bees may prefer key plant species that are more nutritious than others. Is it possible that Japanese honey bees that co-evolved with the surrounding flora have such preferences, whereas western honey bee – alien species that is not connected with this flora via eco-evolutionary interactions – have not such preferences?

*** Agreed, we have included this sound point within the discussion sub-heading “Could Japanese Honeybees be more resilient?”. It reads: “Some studies suggested that honeybees may prefer key plant species because of their ability to perceive higher nutritional value in these plants [74,75]. We suggest that it is possible the JHB may have co-evolved with their native flora to have these advantageous preferences. Whereas the WHB, being both a global generalist and alien species to this lo-cation, is not connected with the local flora via eco-evolutionary interactions and therefore is lacking this foraging and nutritional advantage.”

Please remove from the manuscript superfluous words. Why “immediately” is needed in line 290?

*** Agreed and amended.

Please review studies cited in line 308. The sentence provided in lines 306-308 is not supported by studies 66-68 and studies 9 and 40 are self-citations.

*** We have amended these citations and the sentence they are included with. Notably, the self-citation is appropriate as it remains one of few studies that explicitly look at land use composition and floral forage in honeybees. The other two studies take different approaches, but also link land use with other pollinator forage behaviour. Balfour 2019 was removed.

Last sentence of the manuscript should provide take home message summing-up the current study without referring to any previous studies.

*** Agreed, the final sentence now sums up the findings of the manuscript, without referring to other studies, and reads: “In summary, our study has provided a first-look at the nutrition of A. cerana japonica in its native habitat, which suggests these bees in particular may be resilient to the effects of land use change that have negatively affected other honeybee species elsewhere in the world.”

Round 2

Reviewer 2 Report

I'm OK with this version of the manuscript.

This manuscript is a resubmission of an earlier submission. The following is a list of the peer review reports and author responses from that submission.

Round 1

Reviewer 1 Report

Please see attached comments, copied here for posterity.

Reviewer Comments – 1283573

Summary

This paper attempts to compare the land-use change resiliency of Japonese and western honeybees in the context of a southern-Japan landscape. Much of the comparison struggles with only one species in the mixed-species region being sampled, with the data gathered from colonies exceedingly limited. Further, comparison rests on very poorly quoted reference to literature elsewhere with confusion between nectar and honey foraging / analysis and pollen and bee-bread foraging / analysis.

I appreciate the importance of the hypothesis this study set out to test but for the major reasons I outline below, I do not believe this data able to address these hypothesis and its current presentation is rife with conflation of difference foraging processes and totally lacking in detail or nuance.

This manuscript would need a very significant rewriting with new emphasis on their pollen-diversity data and a much better grasp of citing the literature to be publishable; to address the actual hypotheses they purport to test, a significant amount more data and truly comparative studies would be necessary.

Major Comments

Overall, I find the presentation and ‘comparisons’ of this paper quite confused and muddled, with a lack of clarity of when nectar-foraging and honey production is being discussed, and when pollen foraging and bee-bread production is being discussed.

The authors only present results from samples from Japanese honeybees but say that western honeybees are also kept in this area: a comparative study collecting data from both types of colonies in the same environment would allow them to make direct comparisons. Currently, much of the comparative speculation conflates socio-geographic differences between Europe & North America with Japan and the possible biological differences between Japanese and Western honeybees. There is no clear line of argument about whether this an effect of: the Japanese environment, the Japanese Honeybees, or an interaction between the two. A comparative study between both species in the same environments is necessary to answer a lot of the questions the manuscript purports to address.

With respect to the actual data presented, the analyses focus on monosaccharide ratios in honey and dominant pollen biotypes in pollen either trapped at entrance, directly sampled from corbicula, or taken from in-comb bee-bread. The lack of change in monosaccharide ratios is presented as some measure of resilience but no evidence, putative mechanism, or published literature relating this metric to western honeybee health. Further, the manuscript’s focus (as detailed below) on nectar foraging repeatedly mis-cites pollen-foraging papers. There is little commentary from the manuscript on their pollen-collection results, which would be the far more pertinent side of their data to present in the context of the literature cited. This conflation of two very different foraging modes is severely misleading and undermines the work.

Further, the reliance on sugar and diversity analysis means nothing is said about actual honey or pollen volumes, brood patterns or colony sizes, infectious disease burdens or any other metric of colony strength. The absence of this data again severely inhibits any ability to draw from this work whether land-use is impacting Japonese honeybee population health in this context.

L262-270: Comparisons to A. mellifera around land use impacts are often very vague and not explicit in their suggested mechanisms. How does urban vs rural land use change monosaccharide ratios in honey in the western species, and does this matter? Have explicit links been made relating glucose concentration to honeybee health? One might expect hydrogen peroxide to change with altered glucose/fructose ratios, although recent research one this dismisses that as a non-dominant factor in determining antimicrobial competency of honey. What, then, exactly, are the authors trying to say about these land-use monosaccharide correlations?  Both of the Dr Donkersley’s cited papers [8,9] – which are excellent pieces of research – discuss pollen provision, rather than nectar.

L283-285: Again, we see conflation of nectar and pollen resources, with the authors quoting that nectar was a major part of these studies but that simply doesn’t hold up to inspection. None of these papers discuss nectar deprivation across land-use gradients in honeybees.

L301: Again, this is a principally pollen-foraging paper, yet the authors quote it in the context of nectar foraging. The repeated conflation of nectar/honey and pollen/bee-bread foraging/production is simply not acceptable for publication.

Minor Comments

L85: spelling correction needed, ‘melifera’ -> ‘mellifera’.

L92-94: Incomplete sentence, I understand what the authors are trying to say but currently this doesn’t make technical sense.

L94, L95 & elsewhere: I would recommend the authors move away from using the term ‘fitness’ when centering a manuscript on honeybees, as colony health and productivity are not closely related to reproductive success (i.e. drone matings a successful swarms).

L232-245: Was any correction for multiple testing undertaken here? The repeated tests are increasingly likely to turn up one significant correlation by random chance, my suspicion is there isn’t any real difference at all here even in the 1km / Maltose test. I suggest the authors at least comment on this, or consider where and how they need to correct for multiple testing throughout this manuscript.

L256-259: Journal guidance instructions for authors have been left in the manuscript.

Reviewer 2 Report

This is and a research work on an interesting topic related to the effect of urbanization and agricultural intensification on the managed pollinators and especially the Japanese honey bee Apis cerana japonica. Overall it is well presented but there are some points that the Authors need to clarify or elaborate further in the presentation of results and discussion. Please see attached file.

Reviewer 3 Report

Research Article „Japanese honeybees (Apis cerana japonica Radoszkowski, 1877) may be resilient to land use change” written by Philip Donkersley, Lucy Covell and Takahiro Ota has scientific value and meets the aims and scope of the Insects journal, however I cannot recommend for publication the current form of the manuscript due to the reasons given below. The current version of the manuscript has many flaws but also many qualities. Although flaws are not critical (some of them are low, some are moderate and a few are important), their considerable number makes the current version of the manuscript scientifically inadequate. Therefore, I propose to reject the manuscript and give the Authors time needed to rewrite the manuscript. At the same time, I encourage the Authors to re-submit corrected version of the manuscript. To help the Authors improving their manuscript I will now focus on the flaws, however I also emphasize that the study is worth publishing and I thank the Authors for their effort made to perform the study and write this paper.

The current version of the manuscript is written in a bit disorganized manner, it reads like various parts of the manuscript were written by different persons without communication between them. Wording is inconsistent, especially term “honeybee” is unclear and used sometimes referring only to Apis mellifera, sometimes referring to various honeybees. Some sentences (indicated below) are strange and lacking information needed for full understanding. There is too much so called “handwaving” – i.e. one-sentence strong statements (indicated below) that are not adequately introduced to the Reader. They should be elaborated and substantively clarified to present proper scientific reasoning, and not just the Authors’ opinions, taking advantage of no restriction on the length of the papers. The Authors make reading uncomfortable by not referring to the exact supplementary table or figure (in most cases they refer to “Supplementary materials” but sometimes also to specific table). The Authors did not check the text and files before uploading the manuscript: (1) in the manuscript they refer to the questionnaire in “Supplementary materials” but I was not able to find any questionnaire there, (2) some technical text present in the Insects template was not removed and is mixed with the text of the manuscript. In general, the whole manuscript might be less sloppy, and I hope for more enjoyable reading when reviewing the corrected version. Serious flaw is making equality between managed honeybees and wild pollinators, emphasized in Simple Summary, Abstract and Introduction. Managed honeybees are part of human agriculture, like chickens and cows, and they are not “threatened globally”, as the Authors suggest. However, they play significant role in human food production, and are used on a huge scale as artificial (not natural) pollinators of crops and to produce honey, beeswax and propolis. These are two different things (1 – wild pollinators threatened globally, and 2 – managed honeybees that are important for agriculture) and should be clearly introduced and discussed in the manuscript as two separate things. Especially Simple Summary should not be misleading for lay audience. The biggest flaw, however, is methodological, related to statistical analysis of pollen composition. First of all, I don’t fully understand the method related to pollen analysis, because in lines 130-131 the Authors have written that two types of hives were utilized: 2 urban and 2 rural but this division is not visible in the Results section – it seems that all 4 hives were analyzed together. Secondly, and more importantly, regardless if 2 environments x 2 hives or 4 hives altogether were used, this is inadequate number of replications to make reliable any inferential statistical analysis. The Authors should instead provide simple descriptive statistics (mean, maximum, minimum, etc.) without computing any tests.

Saying this I must admit that the study provides new and interesting knowledge on understudied honeybee species (Apis cerana), and again I thank the Authors for their effort undertaken to study this bee. I will certainly recommend for publication appropriately corrected version of the manuscript.

Below I present detailed comments.

Simple Summary – lines 8 – 19 – in its current form it is not suitable for lay audience. Please rewrite it considering that this is not another version of the abstract, but this is popular-science summary of your study. Please do not mislead your readers providing strong and short oversimplified sentences related to pollinators. Please clearly write what does “honeybee” mean, what is the difference between western honeybee and Japanese honeybee and what species actually was studied here. Please do not use scientific jargon here, like e.g., “species of honeybee that occurs within its natural host range”.  

Abstract

Lines 20 – 23: two issues are mixed here: wild pollinators being threatened globally and managed honeybees that we use to produce food. Please do not lump together completely different realities: ecological reality (functioning of ecosystems), biological conservation reality (conserving wild species) and agricultural reality (food production).

Lines 25 – 32: it is not clear what species was studied here and after reading this abstract, I thought that two species of honeybees were studied. Please clarify.

Lines 33 – 37: this should be appropriately corrected after rewriting Discussion (comments below).

Introduction

There are many short paragraphs (2-3 sentences) disconnected logically from each other. This makes the manuscript laborious to read. I don’t see clear line of thoughts in the Introduction, it is chaotic and ambiguous. Please improve the clarity, flow, and general presentation of the Introduction to allow the Reader understanding of your scientific questions and their importance.

Lines 42 – 44: I don’t understand what background information the Authors wanted to provide here. Please elaborate and clarify.

Lines 48 – 54: here and further in the text when discussing similar issues, only self-citations are provided as references. In all cases, where only self-citations are given, other studies are also available, providing richer context, different angles of view and more deep knowledge as different authors use various approaches, study systems, and frameworks. Please make advantage of all the available science, not only your own studies that are great but not provide full picture.

Considering this particular paragraph, I suggest providing some additional details allowing for better understanding of what the Authors wanted to say. In its current form the paragraph is partial and too vague.

Line 55: this sentence and provided references relate rather to wild bees and other wild pollinators than to managed honeybees. Please clarify this issue.

Lines 68 – 69: this sentence is disconnected from the rest of the text. Why does it appear here? The Authors should clarify what they wanted to say.

Lines 81-83: this sentence does not provide any context since it is hard to compare Japan with England without information on the area and population density. Additionally, it may be more justified to compare beekeeping in Japan with beekeeping in countries of continental Asia or general mean values for Asia.

Lines 84 – 90: I don’t understand this paragraph. I suggest providing here reader-friendly description of what the Authors actually want to emphasize, using the Authors’ own words rather than citing beekeepers.

Lines 92 – 94: I don’t understand this sentence. It looks like something has been cut.

Lines 97 – 108: I suggest changing the subtitle of this paragraph, since it is not related to bee forage ecology. Instead, the Authors presented here the main aim and scope of the study.

Materials and Methods

Lines 111 – 112: please do not use imprecise wording and vague pronouns in the manuscript here and further. Here, instead of “various dates” the Authors should provide precise dates.

Line 114: please be kind to your Reader. It is better to refer to specific table or figure than to “Supplementary Materials” containing a lot of stuff. In this particular case I was not able to find a questionnaire in provided supplement. Please attach needed questionnaire.

Lines 119 – 123: please remove „briefly”

Lines 130 – 131: the Authors provide here information that 2 urban hives and 2 rural hives were used but this 2 X 2 study system is not reflected in analysis and in results provided by the Authors. Please clarify what actually was done and how the data were analyzed. Please also refer to the fact that extremely small number of replications (either 2 X 2 hives or 4 hives – both are extremely small) does not allow for performing any statistical test. This should be clearly acknowledged by Authors in the corrected version of the manuscript. Instead, descriptive statistics may be provided. Please also read my comment below, related to the “Results”.    

Lines 141 – 142: please refer to specific table/figure.

Figure 1: please make numbers easily readable.

Results

Lines 188 – 231: part of the manuscript related to pollen analysis should be completely rewritten, acknowledging consequences of the use of extremely small number of repetitions. Please remove statistical tests and instead provide only simple descriptive statistics (mean, maximum, minimum, etc.) and narrative description (e.g., X was 2-times higher than Y, in month Z the pollen Q was especially rich in our samples contributing to E % of all pollen grains… et. ) without testing any statistical differences. It is OK to provide such information if the data collected do not allow for doing substantially correct statistical analysis. It is unacceptable to create artificially statistically significant results when collected data do not allow for performing analyses.

Discussion

In general, the “Discussion” is oversimplified and vague. Instead of discussing obtained results, the Authors provided some strong statements. Background justifying the statements is not given. I suggest elaborating more on the issues provided in discussion and to focus on providing gripping and coherent story. During rewriting, please focus on questions: what important story, interesting for the scientific community, can be presented based on the obtained results? How Apis cerana may be central point of this story?

Lines 256 – 259: please remove these sentences.

Line 260: fructose “or” glucose? Is it correct? In my opinion it should be fructose “and” glucose

Lines 266: only autocitations here. Please consider my previous comment on autocitaions.

Line 268: pollen was not analyzed “from a substantial number of sites”. Please provide number of sites from which pollen was collected.

Lines 268 – 269: „suggest that Japanese honeybees in Japan may be less susceptible to the negative effects of urbanization” - the Authors should elaborate much more on this issue. In “Discussion” section of a manuscript the Authors should not give short, strong, and undiscussed statements like this one. Instead, background for forming such statements should be provided via discussing obtained results referring to the literature.

Line 285: only autocitations.

Lines 289 – 290: I don’t understand why and how “the social construct of what constitutes a “city” in Japan is observably distinct to other sites”. Please elaborate.

Lines 299 – 302: either “equally possible” or “more likely”, please be consistent.

Conclusions

Lines 328-333: Please clarify this paragraph. It is hard to understand what the Authors wanted to say.